# Migratory allylic arylation of 1,n-enols enabled by nickel catalysis

**Dan Zhao[1,3], Bing Xu[1,2,3] & Can Zhu [1] ✉**

Transition-metal-catalyzed allylic substitution reactions (Tsuji−Trost reactions) proceeding via a π-allyl metal intermediate have been demonstrated as a powerful tool in synthetic chemistry. Herein, we disclose an unprecedented π-allyl metal species migration, walking on the carbon chain involving 1,4-hydride shift as confirmed by deuterium labeling experiments. This migratory allylic arylation can be realized under dual catalysis of nickel and lanthanide triflate, a Lewis acid. Olefin migration has been observed to preferentially occur with the substrate of 1,n-enols ($n \geq 3$). The robust nature of the allylic substitution strategy is reflected by a broad scope of substrates with the control of regio- and stereoselectivity. DFT studies suggest that π-allyl metal species migration consists of the sequential β-H elimination and migratory insertion, with diene not being allowed to release from the metal center before producing a new π-allyl nickel species.

Transition-metal-catalyzed allylic substitution reactions (Tsuji−Trost reactions) have emerged as a powerful tool for the construction of carbon−carbon and carbon−heteroatom bonds with a broad scope of both electrophiles and nucleophiles (Fig. 1a)[1]. By employing an appropriate ligand, the utilization of a transition metal catalyst, including Pd[2–5], Ir[6,7], Rh[8–14], Ru[15–20], Co[21–23], Cu[24–32], Mo[33–41], W[42], and Ni[43–60] realizes the control of regio- and stereoselectivity during nucleophilic substitution of the key π-allyl metal intermediate (**Int-A**), thus to deliver regio- and/or stereomerically pure organic compounds. On the other hand, migratory coupling reactions, combining bond migration with a coupling process, display unique regioselectivity controls, e.g., multiple 1,2-hydride shifts result in the walking of carbon−metal bond over long distances via olefin-metal intermediate (olefin-[M]-**Int**), leading to remote selectivity in coupling reactions (Fig. 1b)[61–70]. Recently, the groups of Kawatsura[54] and Stanley[57] independently reported nickel-catalyzed arylative substitution of homoallylic carbonates and alcohols respectively, and 1,3-hydride shift was involved during olefin migrations. In contrast, migratory allylic substitution, in which the walking of π-allyl metal species proceeds over the carbon chain via multiple 1,4-hydride shifts, are still unexplored (Fig. 1c). During one migration cycle, sequential β-H elimination and migratory insertion occurs with diene-metal complex (diene-[M]-**Int**) as the key intermediate[71,72].

Herein, we disclose our recent observations on the migratory dehydroxylative allylic arylation under the catalysis of nickel and lanthanide triflate, a Lewis acid (Fig. 1d).

## Results and discussion
### Optimization of the reaction conditions
Preliminary attempt began with the substitution reaction of allylic alcohol **1a** with PhB(OH)$_2$ (**2a**) under the catalysis of Ni(cod)$_2$. To our surprise, when electron-rich ligand PCy$_3$ was employed, the reaction does not undergo a direct allylic phenylation pathway to access **3-A** or **3-B**[73]. Instead, migratory phenylation product **3** was unexpectedly obtained in 30% yield (Table 1, entry 1), meanwhile the formation of ketone **4** was also detected, generated from **1a** via intramolecular transfer hydrogenation. Obviously, migration of allylic species occurs preferentially, and product **3** was consequently produced via coupling reaction with PhB(OH)$_2$ as the last step. Interestingly, base does not favor the Suzuki coupling reaction towards **3** (Table 1, entry 2). Therefore, a series of Lewis acids were investigated as additive to the reaction respectively (Table 1, entries 3−9). To our delight, La(OTf)$_3$ gave the best performance, and the yield was dramatically improved

[1]Department of Chemistry, Fudan University, 2005 Songhu Road, Shanghai 200438, China. [2]Zhuhai Fudan Innovation Institute, Zhuhai 519000, China. [3]These authors contributed equally: Dan Zhao, Bing Xu. ✉e-mail: zhucan@fudan.edu.cn

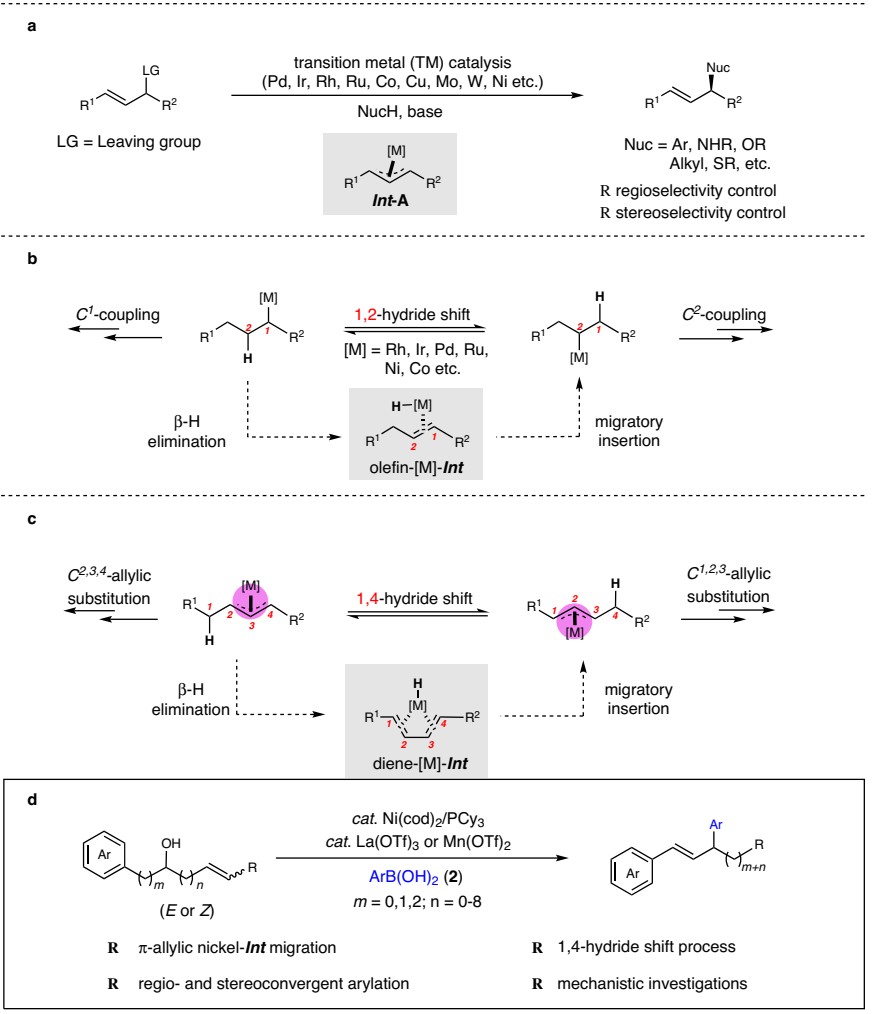

**Fig. 1 | Migratory Allylic Substitution Reactions. a** Classic allylic substitution reactions. **b** Migratory coupling reaction via olefin intermediate (well developed). **c** Migratory allylic substitution reaction via π-allyl metal migration (unexplored). **d** This work.

to 68%, with side product **4** less than 5% (Table 1, entry 9). Further ligand screening fails to give better results (Table 1, entries 10–14). It is noteworthy to mention that bidentate ligands completely prevent the formation of **3**. The yield of migratory arylation was improved to 75% in the solvent of methyl *tert*-butyl ether (MTBE) (Table 1, entries 15–19). Increasing the temperature to 110 °C does not give further promotion (Table 1, entry 20). Finally, Ni(cod)$_2$ (10 mol%), PCy$_3$ (20 mol%), La(OTf)$_3$ (20 mol%), and PhB(OH)$_2$ (2.0 equiv) in MTBE at 90 °C for 12 h were defined as the optimal reaction conditions for additional studies.

**Scope of the reaction**
With the optimized conditions in hand, we started to investigate the scope of this coupling reaction under dual catalysis (Fig. 2). Aryl-boronic acids irrespectively bearing electron donating groups or electron withdrawing groups all reacted with good yields (**3**, and **5**–**10**). To our delight, a variety of functional groups, including 2-Me, 2-OMe, 2-F, 3-Me, 3-OMe, 3-F, 3-CF$_3$, 4-Me, 4-$^t$Bu, 4-Ph, 4-OMe, 4-F, 4-Cl, 4-CF$_3$, and 4-OCF$_3$ on the benzene ring turned out to be compatible under the standard reaction conditions (**11**–**25**). Notably, medicinally prevalent aromatic moieties other than benzene were incorporated smoothly, including 1-naphthyl, 2-naphthyl, 3-indolyl, 2-thiophenyl, and 3-thiophenyl groups (**26**–**30**). 1,2-Enol featuring a terminal olefin works as well, affording arylated product **31** in 63% yield correspondingly. Scope of 1,2-enols could also be extended to those bearing long ali-phatic chains (**32**–**35**). However, migratory phenylation fails to

construct a quaternary carbon center in **36**. Moreover, 1,2-enol fea-turing a tertiary alcohol led to tri-substituted alkene via migratory arylation process. The reactions of cyclic 1,2-enols work nicely to afford indene derivative **37**, bearing a fused five-membered ring, as well as product owning a fused six- (**38**) or seven-membered ring (**39**), and such units have been widely found in various natural products and drug molecules. Finally, it is noteworthy to mention that all the reac-tions give regio- and stereospecific products via migratory allylic substitutions, exhibiting the synthetic robustness of this dual catalytic protocol.

To learn more about reactivity of the migratory allylic arylation transformation, comparison experiments were carried out under the standard conditions by employing (*E*)-enol and its stereoisomer (*Z*)-enol parallelly. The reaction of (*E*)−**1a** afforded migratory phenylated product **3** in 72% yield, and almost the same yield (73%) of **3** could be produced starting from its stereoisomer, (*Z*)−**1a** (Fig. 3a). Similar results could be obtained with tertiary enol (*E*)-**1b** or (*Z*)-**1b** as sub-strates, yielding 59% and 58% of the same product **37** respectively. These outcomes suggest the reaction proceeding via the same intermediate from both stereoisomers, implying the intermediacy of π-allyl metal species. Next, the scope of enols was investigated with OH group and/or olefin moiety at a different position on the carbon chain (Fig. 3b). 1,3-Enol **1c**, with a terminal olefin could also be employed to yield **3** in 71%, the same product as the reaction of **1a**. 1,4-enol **1d**, with olefin distancing one more carbon from OH group gave a decreased yield of 38%. Direct allylic arylation occurs efficiently with

## Table 1 | Migratory dehydroxylative allylic arylation of 1,2-Enols: condition optimization[a]

| Entry | Solvent | Ligand | additive | Yield of 3 (%)[b] | Yield of 4 (%)[b] | Recovery of 1a (%)[b] |
|---|---|---|---|---|---|---|
| 1 | toluene | PCy$_3$ | — | 30 | <5 | n.d. |
| 2 | toluene | PCy$_3$ | K$_2$CO$_3$ | 19 | <5 | 28 |
| 3 | toluene | PCy$_3$ | FeCl$_3$ | n.d. | 8 | 36 |
| 4 | toluene | PCy$_3$ | Fe(OTf)$_2$ | 60 | 7 | n.d. |
| 5 | toluene | PCy$_3$ | Zn(OTf)$_2$ | 36 | <5 | n.d. |
| 6 | toluene | PCy$_3$ | Mn(OTf)$_2$ | 54 | <5 | n.d. |
| 7 | toluene | PCy$_3$ | In(OTf)$_3$ | 21 | 6 | n.d. |
| 8 | toluene | PCy$_3$ | Sc(OTf)$_3$ | 25 | <5 | n.d. |
| 9 | toluene | PCy$_3$ | La(OTf)$_3$ | 68 | <5 | n.d. |
| 10 | toluene | P$^t$Bu$_3$ | La(OTf)$_3$ | 11 | <5 | 10 |
| 11 | toluene | PPh$_3$ | La(OTf)$_3$ | n.d. | 43 | n.d. |
| 12[c] | toluene | BINAP | La(OTf)$_3$ | n.d. | 20 | n.d. |
| 13[c] | toluene | bpy | La(OTf)$_3$ | n.d. | n.d. | 52 |
| 14[c] | toluene | Phen | La(OTf)$_3$ | n.d. | n.d. | n.d. |
| 15 | dioxane | PCy$_3$ | La(OTf)$_3$ | 52 | 7 | n.d. |
| 16 | DCE | PCy$_3$ | La(OTf)$_3$ | 13 | <5 | 12 |
| 17 | THF | PCy$_3$ | La(OTf)$_3$ | 27 | 6 | 26 |
| 18 | n-hexane | PCy$_3$ | La(OTf)$_3$ | 55 | <5 | n.d. |
| 19 | MTBE | PCy$_3$ | La(OTf)$_3$ | 75(72) | <5 | n.d. |
| 20[d] | MTBE | PCy$_3$ | La(OTf)$_3$ | 75 | <5 | n.d. |

[a]The reaction was carried out with PhB(OH)$_2$ (2.0 equiv.), Ni(cod)$_2$ (10 mol%), Ligand (20 mol%), and **1a** (0.2 M) in indicated solvent (1.0 mL) at 90 °C for 12 h. [b]Determined by $^1$H NMR analysis using dibromomethane as the internal standard, and value in parentheses is the isolated yield. [c]10 mol% of bidentate ligand was used. [d]The reaction was conducted at 110 °C. *n.d.* not detected, *DCE* 1,2-dichloroethane, *THF* tetrahydrofuran, *MTBE* methyl *tert*-butyl ether, bpy: 2,2'-bipyridine, *Phen* 1,10-phenanthroline.

allylic alcohol **1e**. The reaction of allylic alcohol **1 f**, in which two more carbons locate between phenyl and OH group compared with **1e**, resulting in the corresponding phenylated product in a decreased yield of 42%, implying consecutive π-allyl metal species migrations before the final coupling with PhB(OH)$_2$. Further studies were carried out by employing 1,n-enols **1g-m** (*n* = 3–10), featuring different distances of olefin unit from OH group. To our delight, these enols are all compatible in the migratory arylation transformations, leading to the migration products in 25–85% yields. These observations make clear that olefin migration preferentially occurs to generate the corresponding allylic alcohol, which further undergoes a substitution reaction with ArB(OH)$_2$ under nickel catalysis. It is worth noting that, in these types of reactions, the preferential olefin migration undergoes via multiple 1,2-hydride shift, and 1,3-hydride shift was also possibly involved in the reaction of 1,3-enols (e.g., **1 g**)[54]. Given the fact that mixtures of olefin or alcohol isomers are abundant industrial feedstocks, and are more easily accessible with less expense than pure isomers, the direct employment of isomers in regio- and stereo-convergent reactions to produce value-added products is of considerable interest. As shown in Fig. 3c, the reaction using a mixture of six regio- and stereoisomers (1:1:1:1:1:1) delivers regio- and stereo-defined product **3** in 64% yield. Next, when an alkenylboronic acid was employed in the reaction of (*E*)−**1a**, we are glad to obtain the corresponding alkenylated product **3-alkenyl** in 24% yield (Fig. 3d). However, cyclopentylboronic acid fails to promote the alkylation reaction

of (*E*)−**1a**. Finally, asymmetric migratory allylic arylation of enols **1o** and **1 h** was investigated, and preliminary attempts were conducted to achieve good enantioselective controls with chiral bidentate nitrogen ligands **L1** and **L2** respectively (91:9 *er* for **31**, 82:18 *er* for **3**), but with low yields in both reactions (for details see page S57-60 in the supplementary discussion section of Supplementary Information).

### Mechanistic studies

To gain a deeper insight of the mechanism of this migratory arylation reaction, control experiments with enol **1n** were performed. When the reaction was stopped in 1 h, corresponding product **21** was formed in 18% yield, while diene **40** was not detected via dehydration from **1n** (Fig. 4a). Further reaction of diene **40** directly under the standard reaction conditions also failed to yield product **21**. When diene **40** was introduced as an additive in the reaction of **1a**, migratory arylation product **3** was generated from **1a** in 70% yield, while **21** was not detected from diene additive **40**. These observations point out that migratory arylation transformations are not proceeding via simple dehydration of enols, and diene cannot access the final products via hydroarylation, indicating that the intermediacy of diene could be ruled out. Moreover, deuterium labeling experiments were carried out to investigate the reaction details during migration (Fig. 4b). When deuterated $d_2$−**1a** was employed under the standard conditions, 95% deuterium incorporation was observed at 4-position, making clear that a specific 1,4-hydride shift occurs during the migratory process.

**Fig. 2 | Substrate Scope.** The reaction was carried out with enol **1** (0.2 M), ArB(OH)$_2$ (2.0 equiv.), Ni(cod)$_2$ (10 mol%), PCy$_3$ (20 mol%), and La(OTf)$_3$ (20 mol%) in MTBE (1.0 mL) at 90 °C for 12 h.

Finally, no intermolecular H/D exchange was observed from the deuterium cross-over reaction of $d_2$–**1a** and **1e** in a one-pot manner (Fig. 4c), implying the prior formation of the new π-allyl nickel intermediate from diene-Ni-***Int*** before the possible disassociation of diene-ligand from the metal, which differs from many cases involving chain-walking process via olefins[57].

To account for the 1,4-hydride shift process observed by the aforementioned deuterium labeling experiments, density functional theory (DFT) calculations were carried out using **1a** as model substrate along with the Ni(0)−PCy$_3$ catalytic system (for details of the optimized structures see coordination data sets and free energies in Supplementary Data 1). The free energy profiles of the preferred 1,4-hydride transfer pathways for (Z)-**1a** and (E)-**1a** represented by the black line and the blue line, respectively, are shown in Fig. 5, taking the π-allyl nickel species **Int1b** as the free energy reference. In the case of (E)-**1a**, isomerization of π-allyl nickel species **Int1b** affords **Int2b**, which undergoes β-hydride elimination via **Ts1b** with an energy barrier of 20.1 kcal.mol$^{-1}$ (**TS1b** with respective to **int1b**) to form the diene-nickel complex **Int3b**. Hydrometallation of **Int4b**, generated by the further isomerization of **Int3b**, needs to overcome an activation barrier of only 5.1 kcal.mol$^{-1}$ (**Ts2b**), providing a new and desired π-allyl nickel species **Int5**. A similar situation can be found in the case of (Z)-**1a** with slightly higher free energy demanding. These calculations indicate that both stereoisomers of **1a** can undergo the 1,4-hydride transfer

process to furnish the same π-allyl nickel complex **Int5**, which is in line with the experimental observation in Fig. 3. Moreover, the dissociation of the diene-nickel complex **Int3a** to form **Int3a-p1** and **Int3a-p2** indicated by red line was also considered, and the higher energy barrier rules out the possibility of the formation of diene, consistent with the results experimentally observed in Fig. 4.

Based on the regiochemical outcome, mechanistic experiments in Fig. 4, and DFT calculation results in Fig. 5, a possible mechanism for the migratory allylic arylation reaction is proposed in Fig. 6. Ligand exchange of Ni(cod)$_2$ with PCy$_3$ generates the active catalyst Ni(PCy$_3$)$_2$, which undergoes the reaction with allylic alcohol **1a** via oxidative addition to give π-allyl nickel species **Int1**. Olefin migration occurs to produce the corresponding allylic alcohols when 1,n-enols (n ≥ 3) are employed, e.g., 1,3-enol **1c**. The hydroxyl group performs as a leaving group after being activated by La(OTf)$_3$, a Lewis acid. β-Hydride elimination of **Int1** affords diene-nickel complex **Int3**, which further transforms to a new π-allyl nickel species **Int5** via hydrometallation. It is noteworthy to mention that the active NiH species in **Int3** is believed to be generated via β-hydride elimination, instead of being formed via oxidative addition of alcohols in many other cases[74–77]. Transformation from one π-allyl species (**Int1**) to the other (**Int5**) proceeds via an overall 1,4-hydride shift, and disassociation of the diene-ligand in **Int3** from the metal seems not possible as confirmed by the deuterium labeling and cross-over studies. Finally, transmetallation of **Int5** with ArB(OH)$_2$ produces **Int6**, which on subsequent reductive elimination

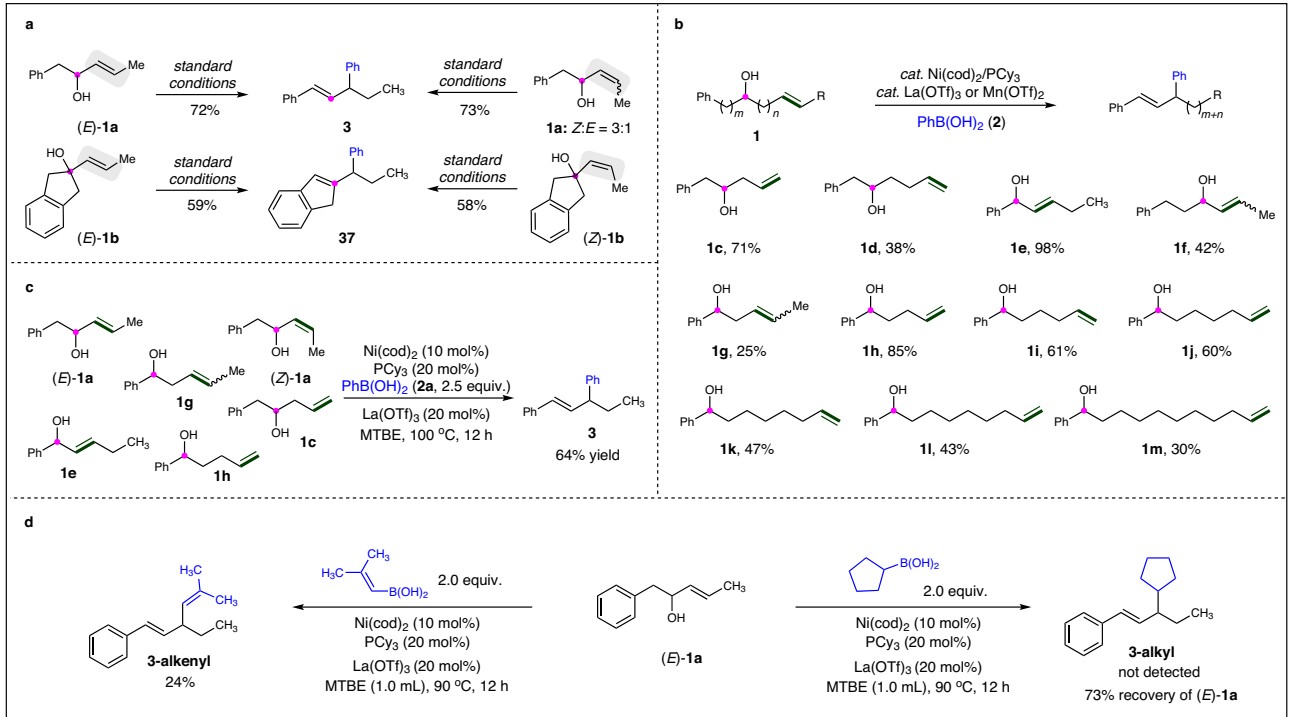

**Fig. 3 | Regio- and stereoconvergent transformations. a** Reactivity comparison of (*E*)- and (*Z*)-enols. **b** Reaction of enols with OH group and/or olefin moiety at different positions. **c** Reaction of a mixture of six regio- and stereoisomers. **d** Reaction of (*E*)-**1a** with an alkenyl or alkyl boronic acid.

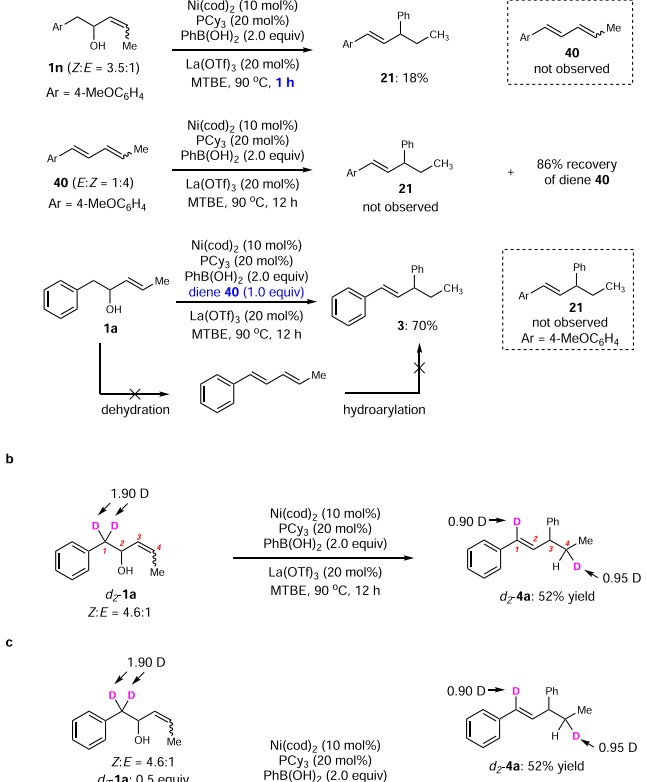

**Fig. 4 | Mechanistic studies. a** Control experiments. **b** Deuterium labeling experiments. **c** Intermolecular cross-over reaction.

leads to the final product **3**, releasing the nickel(0) catalyst to close the cycle.

To sum up, we have developed a migratory dehydroxylative allylic arylation of unactivated 1,n-enols enabled by dual nickel/Lewis acid catalysis with high regio- and stereoselectivity. The reaction proceeds via unprecedented π-allyl nickel species migration involving 1,4-hydride shift as confirmed by deuterium labeling experiments. DFT studies suggest that π-allyl metal species migration consists of the sequential β-H elimination and migratory insertion, with diene not being allowed to release from the metal center before hydrometallation to produce a new π-allyl nickel species. Olefin migration was observed to preferentially occur starting from 1,n-enols (n ≥ 3). The method provides a facile and migratory strategy to construct C(sp²)–C(sp³) bonds with unique selectivity and features a broad substrate scope for both arylboronic acids and 1,n-enols. Further studies on the mechanism, synthetic application, and asymmetric variants are currently ongoing in our laboratory.

## Methods
### Representative procedure for migratory allylic arylation of 1,n-enols enabled by nickel catalysis

Under a nitrogen atmosphere, a mixture of Ni(cod)₂ (5.5 mg, 0.02 mmol), PCy₃ (11.2 mg, 0.04 mmol), Lewis acid (0.04 mmol), and ArB(OH)₂ (0.4 mmol) was added a solution of corresponding enol (0.2 mmol) in MTBE (1.0 mL). The reaction was sealed and stirred at 90 °C or 100 °C for 12 h. Subsequently, the reaction was cooled down to room temperature and the mixture was evaporated and purified via column chromatography on silica gel (eluent: petroleum ether/ethyl acetate) afforded the desired product.

## Data availability
All data generated or analyzed during this study are included in this Article and the Supplementary Information. Details about materials and methods, experimental procedures, mechanistic studies, characterization data, computational details, NMR and HPLC spectra are

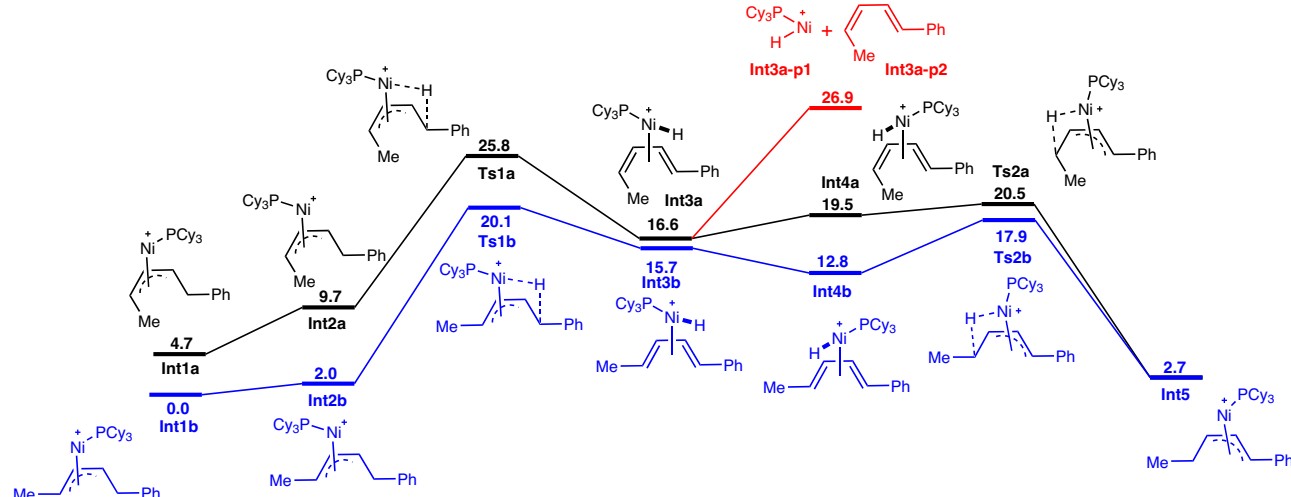

**Fig. 5 | DFT studies.** Free-energy reaction profiles (kcal mol⁻¹) are presented for the 1,4-hydride transfer process of two stereoisomers of **1a**, calculated at the PBE0/combined basis set level at 363 K.

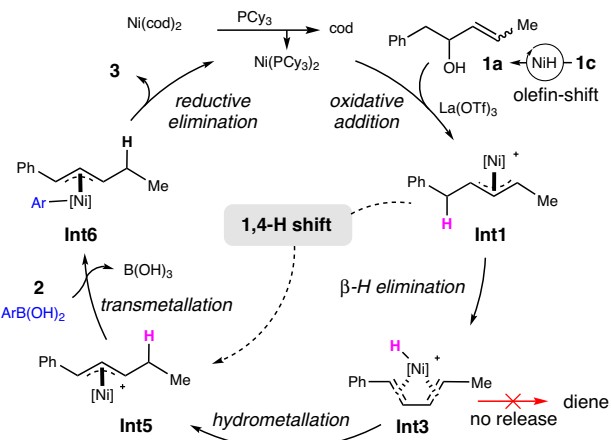

**Fig. 6 | Proposed mechanism.** A plausible reaction pathway for migratory dehydroxylative allylic arylation of 1,n-enols.

available in the Supplementary Information. Calculated coordinates are available in the Supplementary Data file. All other data are available from the corresponding author upon request.

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

## Acknowledgements

We acknowledge the National Natural Science Foundation of China (Grant No. 22271054, C.Z.), the China Postdoctoral Science Foundation (2022M713667, B.X.), the "1000-Youth Talents Plan", and Fudan University (start-up grant) for financial support. We also thank Mr. Jun Zhang for his help in revisions, Prof. Huadong Wang and Bing-Tao Guan for insightful and fruitful discussions. This research paper is dedicated to Prof. Li-Xin Dai on the occasion of his 100th birthday.

## Author contributions

C.Z. conceived the project, analyzed the data, and wrote the manuscript. D.Z. performed the most of experiments. B.X. did the DFT calculations. All authors discussed the results and commented on the manuscript.

## Competing interests

The authors declare no competing interests.
