## [Peer Review File · Nature Communications]

REVIEWER COMMENTS

Reviewer #1 (Remarks to the Author):

The manuscript from Zhu and coworkers describes an interesting Ni-catalyzed hydroarylation of in situ generated 1,3-dienes with arylboronic acids. A broad scope of free 1,n-enols which is easily available are found to be suitable substrates, producing regio- and stereodefined allylic arylation products under mild conditions. In many ways this has been one of the most difficult papers I reviewed recently. Although the overall synthetic results are indeed attractive, I'm a bit unconvinced about the novelty and significance. First of all, it is not suitable to claim that migratory allylic substitution is unexplored (Scheme 1C), as the Ni-catalyzed hydroarylation via in situ generated 1,3-dienes from homoallylic alcohols was already reported by Tsuji and Kawatsura (OL-2020-22-1124, ref 10), Stanley (Chem.Sci.-2022-13-11607, not cited). These two references should be mentioned in the text of introduction and Scheme 1. Second, the current transformation is limited to racemic version. Considering that the asymmetric version of hydroarylation of 1,3-dienes was also realized by Meek (ACIE-2020-59-14070, not cited, monodentate phosphoramidite ligand) and Zhou (CCS Chem-2019-1-328, ref 15b, spiro aminophosphine ligand), the current asymmetric reaction should be possible by using similar chiral ligands. Not in the current form, if the authors could realize the asymmetric version, it would be suitable for Nat. Commun.

Minor comments:

- 1) In Scheme 1, the chain number of diene-[M]-Int is wrong.
- 2) The 1,4-hydride shift is interesting. However, not all the substrates undergo 1,4-hydride shift, it's better to include other types of hydride shift such as 1,3-hydride shift (also interesting).
- 3) Two review papers about Ni-catalyzed hydrofunctionalizations of 1,3-dienes (ACS Catal-2022-12-15638) and hydroarylation (OCF-2022-9-5074) are suggested to be cited. Similarly, a paper of reductive hydroarylation of 1,3-dienes (Chem. – Eur. J. 2021-27-15903) is suggested to be cited.

Reviewer #2 (Remarks to the Author):

Recommendation: Publish after minor revisions noted.

Comments:

The manuscript by Zhu describes a migratory allylic arylation of 1,n-enols enabled by dual nickel/Lewis acid catalysis. This methodology provides a facile and migratory strategy to construct Csp²–Csp³ bonds with high regio- and stereoselectivity and features a broad substrate scope for both aryl boronic acids and 1,n-enols. Notably, the deuterium labeling experiments indicate that the “walking” of π -allyl nickel species involving unprecedented 1,4-hydride shift mechanism. Furthermore, the DFT calculations unveiled sequential β -H elimination and migratory insertion occurs with diene-nickel complex as the key intermediate. The manuscript is well written and the presentation can easily be followed. The experimental section is detailed enough to allow reproduction, the analytics are adequate, and the NMR spectra reveal a satisfying product purity. Based on the above observations, the work is of high quality. This manuscript should deserve to publish in Nature Communications.

Other comments/modifications that should be addressed prior to publication are listed below:

- 1) Owing to migratory phenylation fails to construct a quaternary carbon center ($R_1 \neq H$), this reviewer suggest chemical equation in Scheme 2 should be modified as $R_1 = H$.
- 2) As shown in Scheme 6, olefin-shift (1c to 1a) was promoted by Ni-H. However, authors did not conduct any experiment to explain the formation mechanism of Ni-H. this reviewer suggest corresponding experiments should be presented.
- 3) This reviewer suggest that 2H NMR of d2-1a and d2-4a (using CHCl₃ as solvent) should be measured. This spectrum could better show the dispersion of deuterium.

Reviewer #3 (Remarks to the Author):

In this submitted manuscript, Zhu and co-workers describe a nickel-catalysed migratory allylic arylation of 1,n-enols. Transition-metal-catalysed allylic substitution reactions (Tsuji-Trost reactions) have been demonstrated as a powerful tool in synthetic chemistry. Within the “chain-walking” strategy via 1,4-hydride shifts, the authors realized an unprecedented π -allyl metal species migratory reaction.

The innovation of this manuscript is already evident, given the subsequent in-depth study of the mechanism and the elaboration of critical processes. In particular, the discussion of migration strategy via 1,4-hydride shifts in this manuscript is of further application and reference value. This reviewer believes this manuscript is worth publishing in Nature Communications.

The following minor points should be addressed before publication.

1. Some abbreviations need to be explained in full when they appear for the first time, such as DCE, THF, and others.

2. In the footnote of Scheme 2, PhB(OH)₂ should be written as ArB(OH)₂. In addition, have the authors tried other substrates, such as alkenyl boric acid or alkyl boric acid? Or using aryl halides under reductive conditions?

3. The condition optimization table proves that increasing the reaction temperature would not increase the yield. The standard reaction temperature is 90 °C, while the temperature in Scheme 3c is 100 °C. Is this a clerical error? Alternatively, some explanation should be added.

4. During the synthesis of d₂-phenylacetic acid in Supporting Information, H/D exchange of the carboxylic acid may also occur. Although this intermediate does not affect the final deuterium product, its structure should be revised to 2-phenylacetic-d₂ acid-d.

The following changes have been made:

Reviewer 1:

Question 1: First of all, it is not suitable to claim that migratory allylic substitution is unexplored (Scheme 1C), as the Ni-catalyzed hydroarylation via in situ generated 1,3-dienes from homoallylic alcohols was already reported by Tsuji and Kawatsura (OL-2020-22-1124, ref 10l), Stanley (Chem.Sci.-2022-13-11607, not cited). These two references should be mentioned in the text of introduction and Scheme 1.

Answer: We thank this reviewer for the kind suggestion on the citation of Stanley's report (Chem.Sci.-2022-13-11607), which has been added as ref. 10o. As suggested, these two references have also been included in the text of introduction. However, in both mentioned reports, we can clearly see that only one type of π -allyl nickel was formed from homoallylic carbonates/alcohols in each reaction (see the scheme below).

“Migratory allylic substitution”, which we proposed in scheme 1C (also attached below), is pointing to π -allyl metal species migrating along with the carbon chain. Different types of π -allyl metal species were formed, and can be transformed from one to each other via multiple 1,4-hydride shifts. This type of π -allyl metal species migration was not involved in the reaction mechanism developed by Tsuji and Kawatsura, and Stanley, and it is also unexplored before our discovery. We can also observe consecutive π -allyl metal species migrations (twice) before coupling with $\text{PhB}(\text{OH})_2$, from **1f** in Scheme 3 (also see the scheme below). Specific mechanism of this migration was described by DFT calculation as shown in Scheme 5. To avoid the possible misunderstanding/confusion to readers, we have changed the description in Scheme 1C as “Migratory allylic substitution reaction via π -allyl metal migration

(unexplored)”

Scheme 1C: c) Migratory allylic substitution reaction via π -allyl metal migration (unexplored)

Question 2: Considering that the asymmetric version of hydroarylation of 1,3-dienes was also realized by Meek (ACIE-2020-59-14070, not cited, monodentate phosphoramidite ligand) and Zhou (CCS Chem-2019-1-328, ref 15b, spiro aminophosphine ligand), the current asymmetric reaction should be possible by using similar chiral ligands.

Answer: We thank the reviewer for the kind suggestion. We have added Meek's report as ref. 17d. With enol (*E*)-**1a** as the standard substrate, we tried the monodentate phosphoramidite ligand reported by Meek (ACIE-2020-59-14070), only leading to the racemic product **3** in 12% yield. Moreover, we also applied a chiral spiro-ligand reported by Zhou (CCS Chem-2019-1-328) in this migratory allylic arylation of (*E*)-**1a**, but unfortunately no desired product was detected. Meanwhile, various types of representative and well-known chiral ligands, such as monophosphine ligand (BIDIME), diphosphine ligands (DIOP), ferrocene ligands (BPEphos, Josiphos), axial chiral ligands (BINAP, SYNphos), chiral bis(oxazoline) ligands (BOX), and bis(imidazole) ligands, no enantioselective control could be observed in all attempts (for details, see the scheme below). These results were also enclosed in the SI (S49-52).

The reaction was carried out with (*E*)-1a (0.2 mmol), PhB(OH)₂ (0.4 mmol), Ni(cod)₂ (10 mol %), and ligand (12 mol %) in toluene (1 mL) at 90 °C for 12 h. Yield was determined by ¹H NMR using CH₂Br₂ as the internal standard. Enantiomeric excess (ee) was determined by chiral HPLC. N.D. = not detected. ^aLigand: 20 mol%.

From these results, we conclude that the control of enantioselectivity in this reaction was probably different from that in the hydroarylation of 1,3-dienes reported by Meek and Zhou respectively. Absolute configuration has been determined via the initial chiral recognition via olefin-Ni coordination (Scheme-a below), while chiral recognition in this work proceeds via oxidative addition of C-O bond by Ni to form a chiral π -allyl-Ni species (Scheme-b below).

a) chiral recognition via olefin-Ni coordination

Meek *et al.* *Angew. Chem. Int. Ed.* **2020**, *59*, 14070-14075

Zhou *et al.* *CCS Chem.* **2019**, *1*, 328-334

b) chiral recognition via oxidative addition of C-O bond by Ni

With this analysis in mind, we further investigated asymmetric migratory allylic arylation of enols **1o** and **1h**, given the fact that both reactions start with the enantioselective coordination of olefin to the nickel center, and chiral recognition could be realized by using a suitable chiral ligand. To our delight, after screening over 100 chiral ligands, we found that good enantioselective controls can be achieved with chiral bidentate nitrogen ligands **L1** and **L2** respectively (91:9 *er* for **31**, 82:18 *er* for **3**), but with low yields in both reactions (see the scheme below). We deduce that, in contrast with PCy₃, the catalyst system becomes electron deficient, leading to low catalytic activity towards migratory allylic arylation. These results have been enclosed in the supporting information (S49-52). Further studies on the asymmetric development are currently under way in our laboratory, especially on the yield improvement.

Preliminary attempts for asymmetric migratory allylic arylation

Question 3: In Scheme 1, the chain number of diene-[M]-Int is wrong.

Answer: Yes, the referee is right, and we have updated the chain number of diene-[M]-Int in Scheme 1 accordingly.

Question 4: The 1,4-hydride shift is interesting. However, not all the substrates undergo 1,4-hydride shift, it's better to include other types of hydride shift such as 1,3-hydride shift (also interesting).

Answer: Yes, the referee is right. It is worth noting that these types of reactions do not all undergo the process of 1,4-hydride shift. Substrate scope in Scheme 2 shows the reactivity of enols proceeding via 1,4-hydride shift, while 1,n-enols in Scheme 3b also includes other types of hydride migration processes, such as 1,2-hydride shift, and possible 1,3-hydride shift from 1,3-enols (e.g. **1g**).^{101,100} We have added this description in the manuscript (see the highlighted text above Scheme 3)

Question 5: Two review papers about Ni-catalyzed hydrofunctionalizations of 1,3-dienes (ACS Catal-2022-12-15638) and hydroarylation (OCF-2022-9-5074) are suggested to be cited. Similarly, a paper of reductive hydroarylation of 1,3-dienes (Chem. – Eur. J. 2021-27-15903) is suggested to be cited.

Answer: Thanks a lot for the kind suggestion, and we have added the citation of these three reports as 10p, 10q, and 10r in the article.

Reviewer 2:

Question 1: Owing to migratory phenylation fails to construct a quaternary carbon center ($R1 \neq H$), this reviewer suggest chemical equation in Scheme 2 should be modified as $R1 = H$.

Answer: Thanks a lot for the kind suggestion. We have updated the chemical equation in Scheme 2.

Question 2: As shown in Scheme 6, olefin-shift (1c to 1a) was promoted by Ni-H. However, authors did not conduct any experiment to explain the formation mechanism of Ni-H. this reviewer suggest corresponding experiments should be presented.

Answer: Thanks a lot for the kind suggestion. To confirmed the formation of Ni-H in the reaction, we set up two parallel experiments based on 1,4-enol **1h** and its variant **1h-var**. In the reaction of 1-phenylpent-4-en-1-ol under Ni-catalysis, we can observe obvious olefin migration process as shown by the formation of the corresponding internal olefin in 18% yield in 3 h, increased to 40% in 6 h. However, in the reaction of pent-4-en-1-ylbenzene under the same reaction conditions, such olefin-migration hardly occurs. These outcomes suggest that OH-group could probably be the key element for the formation of NiH species to trigger the subsequent olefin migration, which is consistent with previous report by Zhou *et al.* (*Angew. Chem. Int. Ed.* **2018**, *57*, 461–464), suggesting that NiH species was generated via oxidative addition of an alcohol by Ni(0) (also see the scheme below). These results have been enclosed in the supporting information (S55-56).

1-phenylpent-4-en-1-ol

pent-4-en-1-ylbenzene

Angew. Chem. Int. Ed. **2018**, *57*, 461-464

Question 3: This reviewer suggests that ^2H NMR of $d_2\text{-1a}$ and $d_2\text{-4a}$ (using CHCl_3 as solvent) should be measured. This spectrum could better show the dispersion of deuterium.

Answer: We collected ^2H NMR of $d_2\text{-1a}$ and $d_2\text{-4a}$ (using CHCl_3 as solvent), and the spectra show the dispersion of deuterium. The corresponding data has been updated as well. For the details, see the Supporting Information (S157-158).

Reviewer 3:

Question 1: Some abbreviations need to be explained in full when they appear for the first time, such as DCE, THF, and others.

Answer: As suggested by the referee, we have added explanation of the abbreviations in the first appeared place (for the details, please see the footnote of Table 1).

Question 2: In the footnote of Scheme 2, PhB(OH)_2 should be written as ArB(OH)_2 . In addition, have the authors tried other substrates, such as alkenyl boric acid or alkyl boric acid? Or using aryl halides under reductive conditions?

Answer: Thanks a lot for the kind suggestion. We have modified PhB(OH)_2 as ArB(OH)_2 in the footnote of Scheme 2.

In addition, we have also tried other substrates as suggested by the referee, including alkenyl boric acid and alkyl boric acid. 1) When alkenyl boric acid was employed in the reaction of (*E*)-**1a**, we are glad to obtain the corresponding alkenylated product **3-alkenyl** in 24% yield. However, cyclopentylboronic acid fails to promote the alkylation reaction of (*E*)-**1a**. These outcomes have been enclosed in Scheme 3d. Moreover, we tried coupling reaction using bromobenzene under reductive condition (with Mn or Zn), but no desired product was detected in ether reaction, only with 50% and 47% recovery of enol substrate respectively (for details, see Scheme 3d and S47-49 in the SI).

1) Coupling reaction with alkenyl boric acid

2) Coupling reaction with alkyl boric acid

3) Coupling reaction using bromobenzene under reductive conditions

Question 3: The condition optimization table proves that increasing the reaction temperature would not increase the yield. The standard reaction temperature is 90 °C, while the temperature in Scheme 3c is 100 °C. Is this a clerical error? Alternatively, some explanation should be added.

Answer: Thanks a lot for the careful review. This is not a clerical error. The standard reaction temperature is 90 °C, which is the optimal temperature for substrates proceeding π -allyl metal migrations. For 1,*n*-enols in Scheme 3c, olefin migration preferentially occurs, and this step requires slightly higher temperature, as we used 100 °C therein.

Question 4: During the synthesis of *d*₂-phenylacetic acid in Supporting Information, H/D exchange of the carboxylic acid may also occur. Although this intermediate does

not affect the final deuterium product, its structure should be revised to 2-phenylacetic- d_2 acid- d .

Answer: Yes, the referee is right that H/D exchange of the carboxylic acid also occurs, and we have revised d_2 -phenylacetic acid to 2-phenylacetic- d_2 acid- d . For the details, see the Supporting Information (S21).

REVIEWERS' COMMENTS

Reviewer #1 (Remarks to the Author):

The revised submission by Zhu and co-workers represents a significantly improved version of the original submission. All responses to previous requests are mostly appropriate. Consequently, this paper is now suitable for publication in Nature Communications.

Reviewer #2 (Remarks to the Author):

In the revised manuscript, because the comments raised by the reviewer have been properly answered, the current manuscript can be accepted.

Reviewer #3 (Remarks to the Author):

All my concerns have been well-addressed. Moreover, most of the concerns from Reviewers 1 & 2 have also been addressed.

The authors demonstrated the possibility of the asymmetric migratory allylic arylation of enols after screening over 100 chiral ligands. Good enantioselectivity could be achieved but with low yields.

Although there are imperfections in asymmetric catalysis, this revised manuscript has its merits for publication in Nature Communications.